# A Single Cysteine Residue in the Translocation Pathway of the Mitosomal ADP/ATP Carrier from *Cryptosporidium parvum* Confers a Broad Nucleotide Specificity

**DOI:** 10.3390/ijms21238971

**Published:** 2020-11-26

**Authors:** Martin S. King, Sotiria Tavoulari, Vasiliki Mavridou, Alannah C. King, John Mifsud, Edmund R. S. Kunji

**Affiliations:** Medical Research Council Mitochondrial Biology Unit, The Keith Peters Building, Cambridge Biomedical Campus, Hills Road, Cambridge CB2 0XY, UK; msk@mrc-mbu.cam.ac.uk (M.S.K.); st632@mrc-mbu.cam.ac.uk (S.T.); vm363@mrc-mbu.cam.ac.uk (V.M.); ak2022@mrc-mbu.cam.ac.uk (A.C.K.); John.Mifsud@cantab.net (J.M.)

**Keywords:** substrate specificity and selectivity, substrate binding site, adenine nucleotide translocase, adenine nucleotide translocator, SLC25 mitochondrial carrier family

## Abstract

*Cryptosporidium**parvum* is a clinically important eukaryotic parasite that causes the disease cryptosporidiosis, which manifests with gastroenteritis-like symptoms. The protist has mitosomes, which are organelles of mitochondrial origin that have only been partially characterized. The genome encodes a highly reduced set of transport proteins of the SLC25 mitochondrial carrier family of unknown function. Here, we have studied the transport properties of one member of the *C. parvum* carrier family, demonstrating that it resembles the mitochondrial ADP/ATP carrier of eukaryotes. However, this carrier has a broader substrate specificity for nucleotides, transporting adenosine, thymidine, and uridine di- and triphosphates in contrast to its mitochondrial orthologues, which have a strict substrate specificity for ADP and ATP. Inspection of the putative translocation pathway highlights a cysteine residue, which is a serine in mitochondrial ADP/ATP carriers. When the serine residue is replaced by cysteine or larger hydrophobic residues in the yeast mitochondrial ADP/ATP carrier, the substrate specificity becomes broad, showing that this residue is important for nucleotide base selectivity in ADP/ATP carriers.

## 1. Introduction

*Cryptosporidium* parasites are clinically important parasitic protists that cause cryptosporidiosis, a disease that presents with a gastroenteritis-like syndrome [1]. Among immuno-compromised individuals, cryptosporidiosis is associated with severe, life-threatening illnesses for which there is no effective therapy [2]. *Cryptosporidium* subspecies have been responsible for major outbreaks in the developed world [3,4,5]. Notably, in the developing world, 10–30% of individuals are asymptomatic cyst excretors [6]. In 2004, cryptosporidiosis was included in the World Health Organization (WHO) Neglected Diseases Initiative [7]. Despite its significance for public health, the biology and metabolism of *Cryptosporidium* parasites remain poorly understood. *Cryptosporidium parvum* and *hominis* are the two species responsible for the majority of human infections.

*Cryptosporidium parvum*, in common with a range of anaerobic/microaerophilic eukaryotic lineages, contains a mitochondrion-related organelle, the mitosome [8]. Unlike mitochondria, mitosomes do not contain genetic information, meaning that all of the proteins are encoded by the nuclear genome and imported into the organelle. The metabolic pathways present in mitosomes differ from those in mitochondria. In *C. parvum*, the mitosome does not produce ATP through oxidative phosphorylation, but instead, the organism relies on glycolysis or substrate-level phosphorylation within the mitosome [8,9]. It lacks pyruvate dehydrogenase, most TCA cycle enzymes, and most of the subunits of ATP synthase, except the α and β subunits [8,9].

Despite major metabolic differences, parasitic protozoa are known to contain mitochondrial protein homologues, indicative of a mitochondrial origin of the mitosome. The genome of *C. parvum* has been sequenced [10] and contains eight homologues of the mitochondrial carrier family (SLC25), including a putative orthologue of the mitochondrial ADP/ATP carrier. Mitochondrial carriers are responsible for the transport of metabolites across the mitochondrial inner membrane, linking biochemical pathways in the cytosol with those in the mitochondrial matrix [11,12,13]. In eukaryotic mitochondria, the ADP/ATP carrier replenishes the cell with metabolic energy by importing ADP into the mitochondrion for conversion to ATP and by exporting the synthesized ATP to the cytosol [14].

The ADP/ATP carrier is the best-characterized member of the mitochondrial carrier family (SLC25), and currently the only member for which structural information is available [15,16,17]. Structural and functional analyses have highlighted several important features. The ADP/ATP carrier functions as a monomer [18,19,20,21,22]. It has the tripartite sequence repeats typical of SLC25 members [23] and a three-fold pseudo-symmetrical structure consisting of six transmembrane helices with a translocation pathway through the center of the protein [24]. Each of the three domains consists of two transmembrane helices and a short amphipathic helix in the matrix loop [15]. The odd-numbered helices H1, H3, and H5 have a sharp kink, called the Pro-kink, where a proline residue [15] or serine residue, which mimics proline [16], is located. The proline/serine is the first residue of the signature motif Px[DE]xx[RK].

Two salt bridge networks have been identified, one on the matrix side and one on the cytoplasmic side of the carrier. The matrix salt bridge network, also part of the signature motif Px[DE]xx[RK], forms in the cytoplasmic state, when the substrate binding site is open to the intermembrane space [15,16,25]. The cytoplasmic network forms in the matrix state, when the substrate binding site is open to the mitochondrial matrix [16,26,27,28]. Disruption and formation of these two networks, in an alternating way, leads to changes in accessibility of the substrate binding site to one or the other side of the membrane [26,28]. Glutamine braces stabilize interactions of the matrix network [16], whereas tyrosine braces stabilize interactions of the cytoplasmic network [28]. The tyrosine brace and cytoplasmic network residues are part of another consensus sequence [YF][DE]xx[RK] on the even-numbered helices H2, H4, and H6 [26,28]. Beneath the networks are sets of residues that provide a 15 Å insulation layer [13,28].

The human and bovine isoforms of mitochondrial ADP/ATP carriers are known to have a strict substrate specificity for ADP and ATP, and the deoxy variants [29,30,31,32]. No transport is observed for AMP, purine nucleotides GDP or GTP, or the pyrimidine nucleotides TDP, TTP, CDP, CTP, UDP, and UTP. A single substrate binding site has been located in the central cavity by in silico analyses, by using chemical and distance constraints [33,34], symmetry analysis [26], and molecular dynamics simulations [32,35,36]. The consensus site has three contact points; contact points I and III consist of positively charged residues that bind the phosphate moieties of the nucleotide, whereas contact point II is part of a hydrophobic pocket that binds the adenine moiety [17,33,34]. The residues of this binding site are accessible in the cavity of the matrix and cytoplasmic state [28,37]. However, their role in substrate binding has not been demonstrated directly by experimental or structural work.

Here, we present a detailed functional analysis of the mitosomal ADP/ATP carrier from *Cryptosporidium parvum* (CpAAC). Our results demonstrate that the parasitic carrier can carry out ADP/ATP hetero-exchange and is inhibited by the canonical inhibitors carboxyatractyloside (CATR) and bongkrekic acid (BKA), similar to mitochondrial ADP/ATP carriers. However, it exhibits a unique substrate selectivity profile, which includes thymidine and uridine di- and trinucleotides, in addition to adenine nucleotides. By comparing the sequence of the *C. parvum* homologue with those of mitochondrial ADP/ATP carriers, a cysteine residue is identified, which is in the proximity of contact point II in the proposed binding site and is conserved as serine in mitochondrial orthologues. By mutating the equivalent serine in yeast mitochondrial ADP/ATP carrier Aac2p to cysteine, we have converted a carrier with stringent substrate specificity for ADP and ATP to one that can transport additional nucleotides, matching the substrate specificity profile of the *C. parvum* orthologue. These results advance our understanding of the metabolic processes in *Cryptosporidium* and provide a molecular explanation for the differences in its substrate specificity as compared with mitochondrial ADP/ATP carriers.

## 2. Results

### 2.1. Mitochondrial Carriers of Cryptosporidium parvum

To identify members of the mitochondrial carrier family in *Cryptosporidium parvum,* BLAST searches were carried out using the yeast mitochondrial carriers as templates. The search identified eight putative members on the genome, which all contained the signature motifs Px[DE]xx[RK] and tripartite sequence repeats typical of members of the mitochondrial carrier family (SLC25) [12,13,23,25]. Subsequently, we used two approaches to assign a putative function to these carriers. First, we aligned the identified sequences to the yeast carriers to find the closest homologues. Second, we identified the three contact points [33,34] and asymmetric residues [26] of the proposed substrate binding site to obtain clues about their potential substrates. These analyses identified close homologues of yeast in *C. parvum*, such as a putative mitosomal ADP/ATP carrier (Cgd8_1210, Q5CW91), NAD^+^ or FAD carrier (Cgd1_2370, Q5CSK1), oxoglutarate/malate carrier (partial gene) (Cgd1_600, A3FQC0), phosphate carrier (Cgd2_520, Q5CU30), Mtm1p-like carrier (cgd6_3880, Q5CWS9), Mrs3p-like carrier (Cgd2_1030, A3FQ60), Sam5p or Mme1p-like carrier (cgd6_2350, Q5CX62), and an unknown carrier (cgd6_2360, Q5CX61). Here, we will study, in detail, the putative mitosomal ADP/ATP carrier, called CpAAC, and compare it with mitochondrial ADP/ATP carriers. This comparison is only possible for this member of the mitosomal carrier complement because AAC is the best characterized member of the SLC25 family and the only one for which atomic structures are available [15,16,37].

The importance of ADP/ATP carriers in ATP generation means that most eukaryotes with mitochondria possess at least one, enabling a phylogenetic analysis of a wide range of species. *C. parvum* belongs to the Apicomplexa and the phylogenetic tree based on the sequences reflects this relationship (Figure 1). The mitosomal ADP/ATP carriers of *C. parvum* and *Cryptosporidum ubiquitum* belong to the early branch of the clade of Apicomplexa, which also contains the Hematozoa (e.g., *Plasmodium falciparum*) and Coccidia (*Trypanosoma gondii*).

### 2.2. The Mitosomal ADP/ATP Carrier Retains Structural and Functional Characteristics of the Mitochondrial AAC

The mitosomal carrier has all of the key structural and functional features required for transport, such as the Pro-kink [15,16], matrix salt bridge network [15,16,25], glutamine braces [16], cytoplasmic salt bridge network [16,26,27,28], tyrosine braces [28] (Figure 2 and Figure 3), as well as gate and small residues, required for the conformational state interconversions [12,13,28].

To test its ability to transport nucleotide substrates, we expressed the CpAAC carrier in the cytoplasmic membrane of *Lactococcus lactis*. We have previously shown that this organism is an excellent host for the expression of mitochondrial carrier proteins in a functional state, including the human mitochondrial ADP/ATP carrier HsAAC1 [32,39,40,41,42,43,44,45]. CpAAC was expressed in the cytoplasmic membrane to higher levels than HsAAC1 (Figure 4C), enabling further functional characterization by transport assays (Figure 4).

To determine the transport activity of CpAAC, lactococcal membranes expressing the carrier were fused with liposomes to form fused vesicles. In eukaryotic cells, transport of adenine nucleotides by the mitochondrial ADP/ATP carriers is driven by the chemical gradients of the substrates and the membrane potential, which also determines the directionality [14]. However, the transport activity of these carriers is fully reversible, and they can catalyze both hetero-exchange and homo-exchange of adenine nucleotides in vitro [14,17]. Therefore, to assess transport activity, we extruded the vesicles in the presence of 5 mM unlabeled ADP to incorporate ADP inside the fused vesicles, removed external nucleotide by gel filtration, and initiated homo-exchange with the addition of 1.5 µM [^14^C]-ADP on the outside. CpAAC exhibited high rates of ADP homo-exchange, which could be inhibited by the canonical inhibitors carboxyatractyloside (CATR) and bongkrekic acid (BKA) of mitochondrial ADP/ATP carriers (Figure 4A,B).

The structures of both the bovine and yeast isoforms in the cytoplasmic state reveal the basis of CATR inhibition [15,16]. Of the thirteen residues involved in binding of CATR, ten are strictly conserved in CpAAC (Figure 5A). Residue N104 in ScAac2p, which is replaced by a glycine residue in CpAAC (G99), might form a hydrogen bond with the sulphate moiety of CATR, but it is not consistently observed in the available structures (see [16] for an analysis) (Figure 5A). Two other residues, which are involved in weak van der Waals interactions, have been replaced in CpAAC, but they are unlikely to interfere with binding (Figure 5A). In agreement with these observations, CATR inhibition of ADP homo-exchange by CpAAC is nearly complete (Figure 4B).

Comparison of the model of CpAAC with the structure of the ADP/ATP carrier from *Thermothelomyces thermophila* (TtAac) in complex with bongkrekic acid (BKA) [28,37] shows that all polar interactions are fully conserved, except for N96 in TtAac, which is G99 in CpAAC (Figure 5B). Two residues L135 and S238 in TtAac are involved in van der Waals interactions with BKA, but they are different in CpAAC, being S138 and C238, respectively. Transport by CpAAC is inhibited to 83% by bongkrekic acid, which could be explained by these substitutions interfering to a small extend with the binding of BKA (Figure 4B).

### 2.3. The Mitosomal ADP/ATP Carrier from Cryptosporidium parvum Has a Broad Nucleotide Specificity Profile

Human and bovine ADP/ATP carriers have a strict substrate specificity for ADP, ATP and their deoxy variants [29,30,31,32]. Here, we tested whether the substrate specificity of CpAAC was similar to that of mitochondrial ADP/ATP carriers. We determined the substrate specificity using two methods. First, competition experiments were carried out, in which radiolabeled ADP uptake into vesicles was monitored in the presence of a 1667-fold excess of external non-labeled nucleotides (Figure 6). There was almost complete inhibition of ADP homo-exchange by CpAAC by a broad range of di-and trinucleotides, but not mononucleotides, except for AMP (Figure 6). However, these experiments show that these nucleotides can compete with ADP for binding to some extent, when present in large excess, but they do not demonstrate that these nucleotides are transported by CpAAC.

Second, we performed hetero-exchange experiments with all compounds that effectively competed for ADP uptake to verify whether they were transported by CpAAC (Figure 7A). For this purpose, non-labeled compounds were loaded into fused membrane vesicles, and hetero-exchange was initiated by the addition of radio-labeled ADP and the transport rate was monitored (see inset Figure 7). Only when the internal compound is transported out can uptake of the radiolabeled ADP into the fused vesicles occur. The initial rates of ADP and ATP uptake by CpAAC were 1.71 ± 0.49 and 2.25 ± 1.45 nmol [^14^C]-ADP mg^−1^ total protein min^−1^, respectively, as compared with the rates of TDP, TTP, UDP, and UTP uptake by CpAAC which were 3.15 ± 1.90, 3.62 ± 1.58, 2.52 ± 0.28, and 4.12 ± 2.00 nmol [^14^C]-ADP mg^−1^ total protein min^−1^, respectively. Hetero-exchange was also observed for inosine and cytosine di- and trinucleotides, but at much lower rates (0.76 ± 0.15, 1.21 ± 0.38, 1.04 ± 0.32, and 1.09 ± 0.29 nmol [^14^C]-ADP mg^−1^ total protein min^−1^ for IDP, ITP, CDP, and CTP, respectively). The rate of transport for guanosine nucleotides was low, despite near complete inhibition in the competition assays, suggesting these nucleotides can compete for substrate binding when present in large excess, but they are not transported.

For comparison, we tested the substrate specificity of ScAac2p, for which a structure is available [16], in the same way. Indeed, ScAac2p shares the substrate specificity profile of the human and bovine mitochondrial ADP/ATP carriers, showing the highest rates of transport for ADP, ATP, dADP, and dATP, but no transport for TDP, TTP, UDP, and UTP (Figure 7B). These data show that CpAAC has a broader substrate specificity as compared with ScAac2p and previously characterized orthologues [31,32].

### 2.4. A Single Residue in the Substrate Translocation Pathway Can Broaden the Substrate Selectivity of the Yeast ADP/ATP Carrier

To investigate the molecular basis of the observed substrate specificity of CpAAC, we compared the sequences of ADP/ATP carriers that had a strict specificity for ADP and ATP with the one of *C. parvum* (Figure 2). The only difference in the residues predicted to be near the putative adenine binding site is C238 in *C. parvum*, which is a serine residue in the yeast (S245), bovine (S227) and human (S228) mitochondrial orthologs. In fact, a cysteine residue in that position is only observed in *C. parvum* and other cryptosporidia, and in the mitosomal ADP/ATP carrier of *Entamoeba histolytica*.

Next, we mutated serine 245 of ScAac2p to cysteine, as found in *C. parvum*, to analyze its effect on the substrate specificity of the carrier. To get a more complete picture, we also introduced other substitutions, such as the hydrophobic amino acid residues alanine, isoleucine, leucine, and valine and the polar amino acid residue threonine, which has similar properties as serine. To test the substrate specificity of each mutant variant in hetero-exchange we loaded, in each case, fused membrane vesicles with ATP, CTP, GTP, ITP, TTP, or UTP and initiated transport by the addition of radio-labeled ADP (Figure 8). All mutants were capable of ADP/ATP hetero-exchange; the rates in μmol [^14^C]-ADP mg^−1^ of AAC min^−1^ for S245A (0.47 ± 0.06) and S245C (0.53 ± 0.03) were the same as wild-type (0.53 ± 0.02), the rates for S245I (0.72 ± 0.07) and S245L (0.67 ± 0.13) were slightly higher, and the rates for S245T (0.29 ± 0.02) and S245V (0.37 ± 0.11) were slightly lower than the wild-type. For all mutants, the rates for CTP, GTP, and ITP hetero-exchange were either similar, or slightly lower than the wild-type but overall, very low. However, there was a significant increase in the rate of ADP/TTP and ADP/UTP hetero-exchange for the S245C, S245I, and S245L variants as compared with the wild-type. The rate of ADP/TTP hetero-exchange in µmol [^14^C]-ADP mg^−1^ of AAC min^−1^ for ScAac2p was 0.08 ± 0.02, whereas for S245L, the transport rates were seven-times higher than the wild-type (0.56 ± 0.22), the highest observed. Similarly, the rate of ADP/UTP hetero-exchange in µmol [^14^C]-ADP mg^−1^ of AAC min^−1^ for S245L was more than four-times higher than that of the wild-type (0.45 ± 0.17 for S245L versus 0.10 ± 0.06). Interestingly, the rates of ADP/CTP hetero-exchange for S245C (0.06 ± 0.02), S245I (0.08 ± 0.03) and S245L (0.14 ± 0.05) were not markedly different from the wild-type (0.11 ± 0.02), despite the structural similarity between these pyrimidine nucleotides. These data indicate that the amino acid residue in position 238 in CpAAC and 245 in ScAac2p is critical for determining the substrate specificity of the carrier, suggesting it might have a role in nucleotide base selectivity.

## 3. Discussion

We have used bioinformatics to identify putative mitochondrial carriers encoded by the genome of the parasitic protist *Cryptosporidium parvum*. Whereas most eukaryotes have between 30 and 60 mitochondrial carriers [46], *C. parvum* has only eight, in agreement with the highly reduced functions of the mitosome. A similar reduction in the mitochondrial carrier complement has been observed previously for other eukaryotic parasites, which occurred in adaptation to their host cell exploitation [43,47]. One potential reason is that parasites can acquire nutrients from the host cells, meaning that many synthetic pathways and accompanying transport steps become redundant, leading to gene loss and a more reduced genome.

In this study, we show that the genome of *C. parvum* encodes a mitosomal ADP/ATP carrier protein (CpAAC), phylogenetically related to mitochondrial ADP/ATP carriers, but with a broad substrate specificity. We have previously characterized the substrate specificity of the human mitochondrial ADP/ATP carrier [32], the mitosomal ADP/ATP carrier of the microaerophilic human parasite *Entamoeba histolytica* (EhAAC) [43], and here, the yeast mitochondrial ADP/ATP carrier ScAac2p (Figure 7B). These analyses are consistent with previous studies on ADP/ATP carriers from other species, showing that the substrate specificity is confined to ADP and ATP, and their deoxy variants [29,30,31]. In contrast, CpAAC has gained the ability to transport a broader range of di- and tri-nucleotides, i.e., ADP, ATP, dADP, and dATP, but also TDP, TTP, UDP, and UTP (Figure 7A), which is unprecedented. CpAAC is much more closely related to mitochondrial ADP/ATP carriers with respect to the conserved structural and functional features than the mitosomal carrier of *Entamoeba histolytica* EhAAC [43], which is much more divergent (Figure 2). Although EhAAC also has a cysteine residue in a similar position as C238 of CpAAC, the rest of the putative binding site is not conserved, making it difficult to rationalize the differences in specificity (Figure 2). EhAAC is much smaller in size, has unusually short loops, and has substantial differences in other conserved features of mitochondrial ADP/ATP carriers, such as the networks and braces (Figure 2).

The differences between CpAAC and other characterized mitochondrial ADP/ATP carriers provide new insights into the molecular determinants of substrate binding and transport. One way of explaining these differences is by carrying out docking studies or molecular dynamics simulations, but there is a fundamental problem. We have analyzed the energetics of the transport cycle by treating the ADP/ATP carrier as a nanomachine that moves stochastically and continuously from the matrix to cytoplasmic conformation under the influence of thermal energy [48]. This analysis shows that the tightest binding of the substrate occurs in the occluded state [48], which is a defined state [49], but its structure has not been determined.

Still, it should be possible to compare the properties of the substrate binding site with those of the substrates, in particular, the nucleotide bases. We had previously proposed a putative binding site using a novel binding site search procedure [33,34], which has been confirmed by other bioinformatic studies and molecular dynamics simulations [26,32,35,36]. The proposed binding site in the yeast ADP/ATP carrier ScAac2p consists of positively charged residues K38, R96, and R294, which may form ionic interactions with the phosphate moieties of ADP and ATP [33,34] (Figure 9A). It contains a tyrosine residue Y203, which may form aromatic stacking arrangements with the adenine ring [32,33,34,35]. Residue Y203 is flanked by residues G199, I200, which have been implicated in substrate binding previously [33,34], as well as S245, and S242, which are in the vicinity (Figure 9A). Together, these five residues form a fairly large hydrophobic pocket, which could function as an adenine binding site (Figure 9A). What about the properties of the substrates? Electrostatic surface calculations of the nucleotide bases have shown that cytosine, guanine, and to a lesser extent inosine are quite polar because of strong partial negative charges at the ketone groups [50]. Adenine, thymidine, and uridine have a more even electron distribution and are consequently less polar [50].

What can we learn about the substrate specificity by comparing the properties of the substrate binding sites of the single replacement mutants (Figure 8) and those of the nucleotide bases? First, the nucleotide hetero-exchange experiments clearly show that S245 is not essential for binding of adenine nucleotides, as the replacement mutants can still transport adenine nucleotides with similar rates. This result indicates that the hydroxyl group of S245 is not directly involved in adenine binding, but this residue must still be involved in substrate selectivity, given the specificity broadening observed in CpAAC (Figure 7A).

Second, substitutions introducing a larger and more hydrophobic group than S245 into the putative adenine binding pocket of ScAac2p, such as S245C, S245I, and S245L (Figure 9C–E), enable transport of uridine and thymidine nucleotides (Figure 8). The pyrimidine bases, thymidine and uridine, are smaller than adenosine and, consequently, might fit loosely in the adenine binding pocket of the wild-type yeast protein. The substitutions may make the binding of uridine and thymidine bases possible by a better fit, which still allows adenine to bind, because they are all hydrophobic in nature. Cytosine is also smaller than adenine, but more polar than the other pyrimidine bases, and will not be tolerated well in this hydrophobic site. The purine bases, guanine and inosine, are bigger and more polar than adenine and are not tolerated in the largely hydrophobic binding pocket. The S245T mutation leads to the introduction of a larger polar group in the hydrophobic adenine binding pocket (Figure 9F), and to a reduction in ADP/ATP hetero-exchange (Figure 8), potentially because of less effective binding of the adenine moiety. As expected, this substitution does not promote efficient binding of uridine and thymidine nucleotides. The smaller substitutions, S245V and S245A, do not introduce a significant change in properties of the binding pocket (Figure 9B,G) and their substrate specificities are similar to that of wild-type (Figure 8). Thus, the hydrophobicity and size of the adenine binding site are two key factors that work together to determine the specificity of nucleotide base selection rather than specific interactions.

Overall, our findings provide new insights into the substrate specificity of the ADP/ATP carriers, as well as potential metabolic functions of *Cryptosporidium parvum*. Interestingly, the serine at position 245 is strictly conserved among mitochondrial ADP/ATP carriers. These results may explain why there is a strong selection against mutations at this position. If other nucleotides are transported as well, key processes, such as equimolar exchange of ADP/ATP, as well as nucleotide transport for DNA and RNA synthesis could become deregulated in mitochondria. These processes do not occur in the mitosome of *C. parvum,* and thus there is no selective pressure against broadening of the substrate specificity. Given the absence of oxidative phosphorylation in *C. parvum*, it is likely that CpAAC is responsible for importing ATP into the mitosome for conserved, energy-requiring processes, such as iron-sulphur cluster synthesis [51,52,53]. The broader substrate specificity of the *C. parvum* carrier is intriguing, as the parasite does not have mitochondrial DNA requiring nucleotide import for synthesis and transcription. However, mitosomal function may also require the import of a broad range of nucleotides, given that *C. parvum* relies on salvage pathways for supply of purine and pyrimidine nucleotides [54]. In any case, iron-sulphur cluster synthesis is an essential process in eukaryotes, which requires the import of ATP into the mitosome, and thus CpAAC might be an interesting drug target for treatment of cryptosporidiosis.

## 4. Materials and Methods

### 4.1. In Silico Analysis

To identify the sequences of mitochondrial carriers from *Cryptosporidium parvum*, BLAST searches of the genome sequence stored at the National Centre for Biotechnology Information databases were performed. Protein sequences for the yeast mitochondrial carriers and the ADP/ATP carrier of *Entamoeba histolytica* were used as search templates, using an expected threshold of 1. Identified sequences were aligned with CLUSTALW [55], redundancy was removed by CD-HIT [56], and the sequences were curated manually. BLAST searches of the putative carrier proteins to the yeast genome allowed us to predict their most likely function. ADP/ATP carrier sequences for a range of different eukaryotic species were gathered using NCBI BLAST and Uniprot. Jalview was used to align the sequences. The loop regions of the carriers were removed by using the location of the loops in the known structures (PDB entries 4c9g [16] and 6gci [28]). Once the helical regions were isolated, the alignment was stored as a Phylip file. The phylogenetic tree was generated with RAxML, using a maximum likelihood method [57]. The LG matrix was used with a gamma model of heterogeneity. The tree was visualized using the Interactive Tree of Life browser (https://itol.embl.de). Structural models of CpAAC in the cytoplasmic state and matrix state, based on PDB entries 4c9g [16] and 6gci [28], were calculated by SWISS-MODEL, following the alignment, as shown in Figure 2 [38].

### 4.2. DNA Constructs and Mutagenesis

The *Cryptosporidium parvum* CpAAC gene was codon-optimized, synthesized (GenScript, Piscataway, NJ, USA), and cloned into the *Lactococcus lactis* expression vector pNZ8048 under the control of a nisin A-inducible promoter. The expression strains for the human ADP/ATP carrier HsAAC1 [32] and yeast ADP/ATP carrier [39,41] have been described previously. *L. lactis* strain NZ9000 was transformed with the resulting plasmid by electroporation [45]. Mutagenesis was performed using standard PCR procedures. Plasmids were isolated by miniprep (Qiagen, Hilden, Germany), according to the instructions with one alteration; 10 mg mL^−1^ lysozyme was added to the lysis buffer and the resuspended cells were incubated at 55 °C for 20 min prior to lysis. Gene insertions were confirmed by sequencing.

### 4.3. Growth of Lactococcus lactis and Membrane Isolation

Precultures of *L. lactis* were obtained by inoculating M17 medium supplemented with 1% (*w/v*) glucose and 5 μg mL^−1^ chloramphenicol from glycerol stocks and incubating the cultures overnight at 30 °C with no aeration. Cells were diluted to a starting OD_600_ of 0.1 in fresh medium and grown at 30 °C with no aeration until the OD_600_ reached 0.5. Expression of the recombinant proteins was induced by addition of nisin A with a dilution of 1:10,000 of spent M17 medium from nisin A-excreting *L. lactis* strain NZ9700. Cells were grown for a further 2 h at 30 °C, harvested by centrifugation (6,000× *g*, 10 min, 4 °C), resuspended in 10 mM PIPES pH 7.0, 50 mM NaCl (pH 7) (PIPES buffer), and collected by centrifugation as before. Cells were subsequently resuspended in 50 mL PIPES buffer and disrupted mechanically with a cell disruptor (Constant Cell Disruption Systems, Daventry, UK) at 30 kpsi. Whole cells and debris were removed by centrifugation (10,800× *g*, 15 min, 4 °C) and the membranes were collected by ultracentrifugation (138,000× *g*, 1 h, 4 °C). Pellets were resuspended in PIPES buffer and stored in liquid nitrogen.

### 4.4. SDS-PAGE and Western Blotting

The membrane proteins were separated by SDS-PAGE (4–20% tris-glycine gel, Novex, ThermoFisher, Waltham, MA, USA) and stained using Instant Blue (Expedeon, Cambridge, UK). For Western blotting, the proteins were transferred to PVDF membranes using semi-dry transfer apparatus, and then blocked in buffer containing 5% milk powder and 0.4% Tween 20, for one hour, at room temperature. The membranes were subsequently probed with an anti-ADP/ATP carrier antibody raised in chicken (antigen: YPLDTVRRRMMMT) at 1:20,000 dilution for one hour, followed by an anti-chicken-HRP conjugate at 1:20,000 dilution for another hour. The membrane was developed using Amersham ECL Western blotting detection system (Little Chalfont, Buckinghamshire, UK) for 30 min.

### 4.5. Membrane Vesicle Fusions

*Escherichia coli* polar lipid extract and egg yolk phosphatidylcholine (25 mg mL^−1^ in chloroform from Avanti Polar Lipids, Alabaster, AL, USA) were mixed in a mass ratio of 3:1. The chloroform was evaporated under a stream of nitrogen and the lipids were washed with an equal volume of diethyl ether. When dry, the lipid mixture was resuspended in PIPES buffer with a homogenizer to a final concentration of 20 mg mL^−1^ and frozen in liquid nitrogen. To make membrane fusions, 1 mg *L. lactis* membranes was mixed with 5 mg liposomes (extruded 11 times through a 1 μm polycarbonate filter), diluted to a final volume of 1 mL with PIPES buffer, and fused by seven cycles of freezing in liquid nitrogen and thawing at room temperature, before storage in liquid nitrogen.

### 4.6. Transport Assays

On the day of the experiment, 1 mL of membrane vesicle fusions were thawed, and 100 μL PIPES with 55 mM substrate was added. Vesicles were extruded 11 times through a 1 μm polycarbonate filter, passed through a pre-equilibrated PD10 column to remove external substrate, and collected in PIPES buffer. Transport assays were carried out using a Hamilton MicroLab Star robot (Hamilton Robotics Ltd., Birmingham, UK). Transport of [^14^C]-labeled ADP was initiated by the addition of 100 μL PIPES buffer with 1.5 μM [^14^C]-ADP (2.22 GBq mmol^−1^) to 5 μg vesicles in a MultiScreenHTS-HA 96-well filter plate (pore size 0.45 μm, Millipore, Merck, Darmstadt, Germany), and stopped at 0 s, 10 s, 20 s, 30 s, 45 s, 60 s, 150 s, 5 min, 7.5 min, 10 min, and 15 min by the addition of 200 μL ice-cold PIPES buffer and filtration, followed by two additional wash steps with 200 μL ice-cold PIPES buffer. Levels of radioactivity in the vesicles were measured by the addition of 200 μL MicroScint-20 (Perkin Elmer, Waltham, MA, USA) and by quantifying the amount of radioactivity with a TopCount scintillation counter (Perkin Elmer). Initial rates were determined from the linear part of the uptake curves, typically the first 60 s.

## Figures and Tables

**Figure 1 ijms-21-08971-f001:**
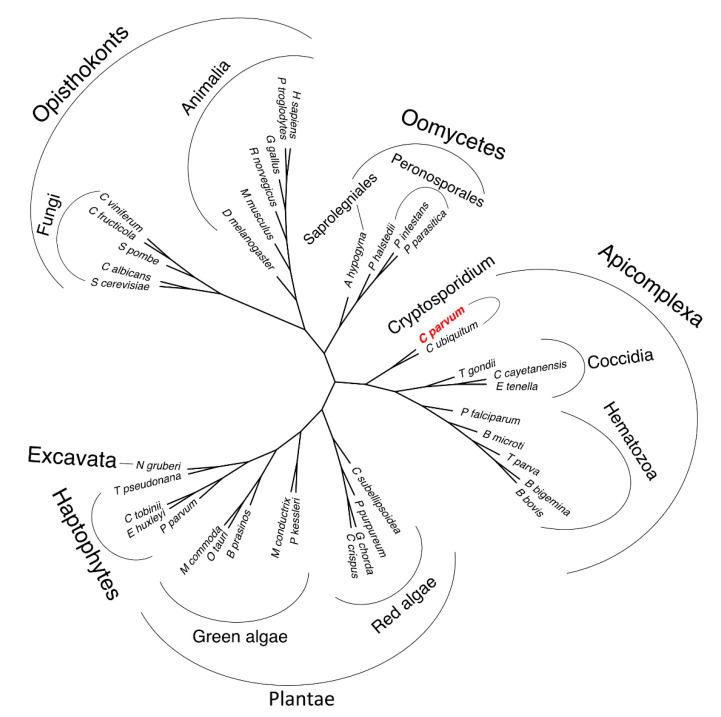
Phylogenetic relationship of mitosomal and mitochondrial ADP/ATP carriers. An unrooted phylogenetic tree of homologous mitochondrial and mitosomal ADP/ATP carrier sequences for a selection of eukaryotic organisms based on alignments. The tree was calculated using RAxML. The mitosomal ADP/ATP carrier of the parasite *Cryptosporidium parvum* is in an early branch of the Apicomplexa, as expected on the basis of species phylogeny.

**Figure 2 ijms-21-08971-f002:**
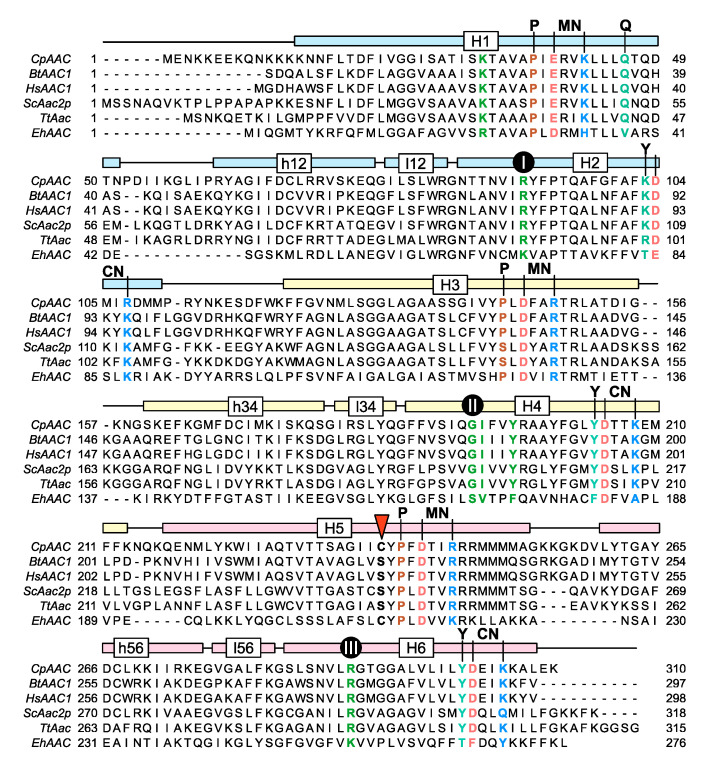
The putative ADP/ATP carrier of *C. parvum* is homologous to mitochondrial ADP/ATP carriers. Alignment of CpAAC with the mitochondrial ADP/ATP carrier of *Bos taurus* (BtAAC1), *Homo sapiens* (HsAAC1), *Saccharomyces cerevisiae* (ScAac2p), *Thermothelomyces thermophila* (TtAac), and *Entamoeba histolytica* (EhAAC). The transmembrane α-helices (H1-6), linker α-helices (l12, l34, l56), and matrix α-helices (h12, h34, h56) are indicated. Repeats 1, 2, and 3 are colored in pastel blue, yellow, and red, respectively. The residues of the Pro-kink (P, brown), matrix network (MN, red and blue), glutamine-brace (Q, cyan), substrate binding site (green), contact points (black circles), tyrosine-brace (Y, cyan), and cytoplasmic network (CN, red and blue) are indicated.

**Figure 3 ijms-21-08971-f003:**
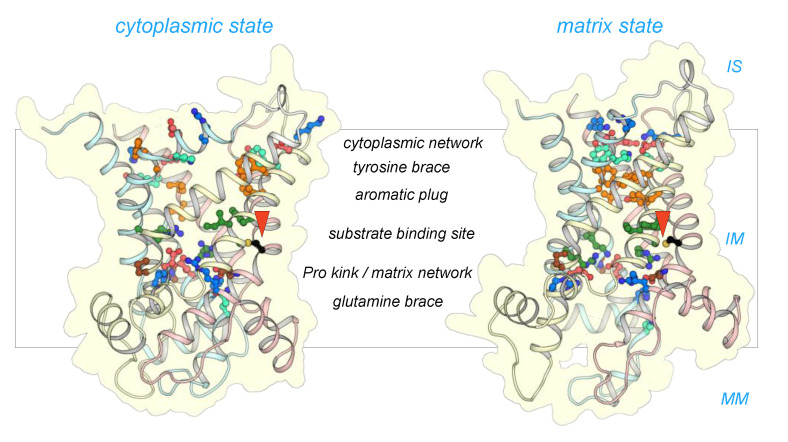
CpAAC has all of the key features of a functional ADP/ATP. Structural models of CpAAC, in the cytoplasmic state (**left**) and matrix state (**right**), based on Protein Data Bank (PDB) entries 4c9g [16] and 6gci [28] respectively, calculated by SWISS-MODEL [38]. Repeats 1, 2, and 3 are colored in pastel blue, yellow, and red, respectively. The residues of the Pro-kink (brown), matrix network (red and blue), glutamine-brace (cyan), substrate binding site (green), aromatic plug (orange), tyrosine-brace (cyan), and cytoplasmic network (red and blue) are indicated. Residue C238 in CpAAC is indicated by a red arrowhead. IS, intermembrane space; IM, mitochondrial inner membrane; MM, mitochondrial matrix.

**Figure 4 ijms-21-08971-f004:**
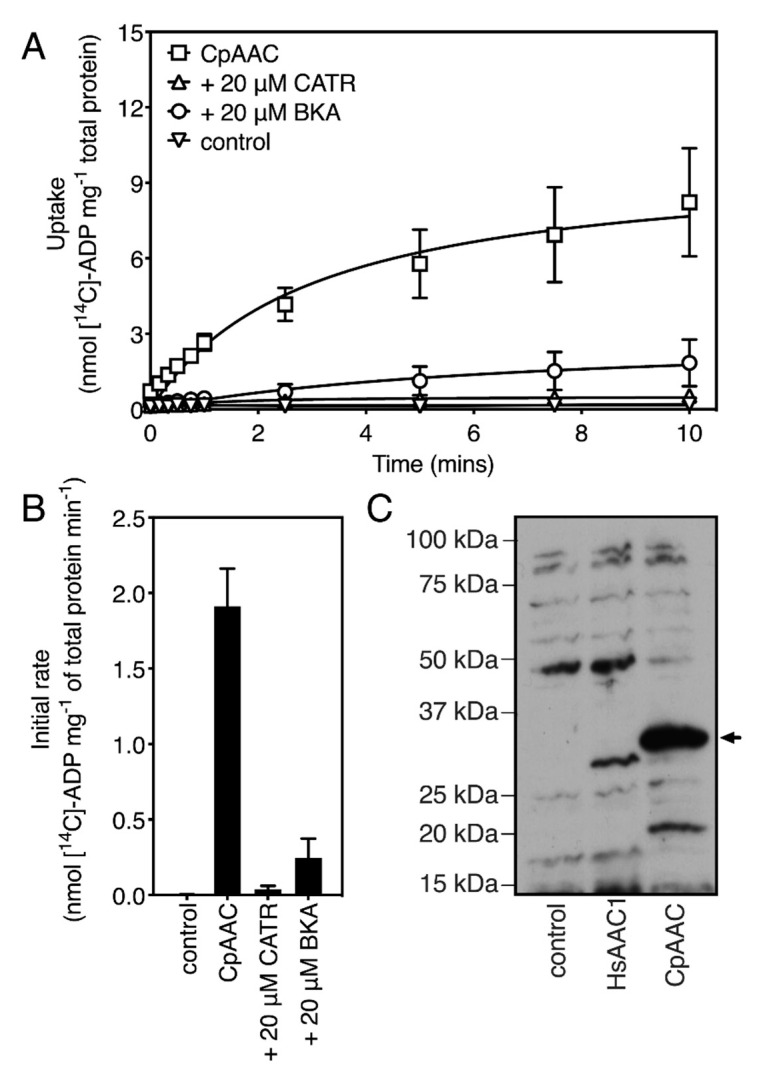
CpAAC expressed in *Lactococcus lactis* retains functional characteristics of the mitochondrial ADP/ATP carriers. (**A**) ADP transport by CpAAC can be inhibited by carboxyatractyloside (CATR) and bongkrekic acid (BKA), the canonical inhibitors of mitochondrial AACs. Fused membrane vesicles of lactococcal membranes expressing CpAAC were preloaded with 5 mM ADP in the absence (squares) or presence of 20 μM CATR (triangles) or BKA (circles). The empty vector controls are shown as inverted triangles. Transport was initiated by the addition of 1.5 μM [^14^C]-ADP; (**B**) The specific initial transport rates, when the applied chemical gradients of radio-labeled and cold substrates are maximal, are derived from the linear parts of the uptake curves, typically in the first minute, corrected for the amount of protein. The data are represented by the mean and standard deviation of two biological repeats, each performed in quadruplicate; (**C**) Western blot of cytoplasmic membranes of *L. lactis* strains expressing CpAAC or human AAC1 (HsAAC1), which was characterized previously by [32], or transformed with empty vector. The band of CpAAC (approximately 33 kDa) was detected with a chicken antibody raised against the antigen YPLDTVRRRMMMT and anti-chicken-horseradish peroxidase conjugate. It is marked with an arrow. 10 µg of total protein was loaded per lane.

**Figure 5 ijms-21-08971-f005:**
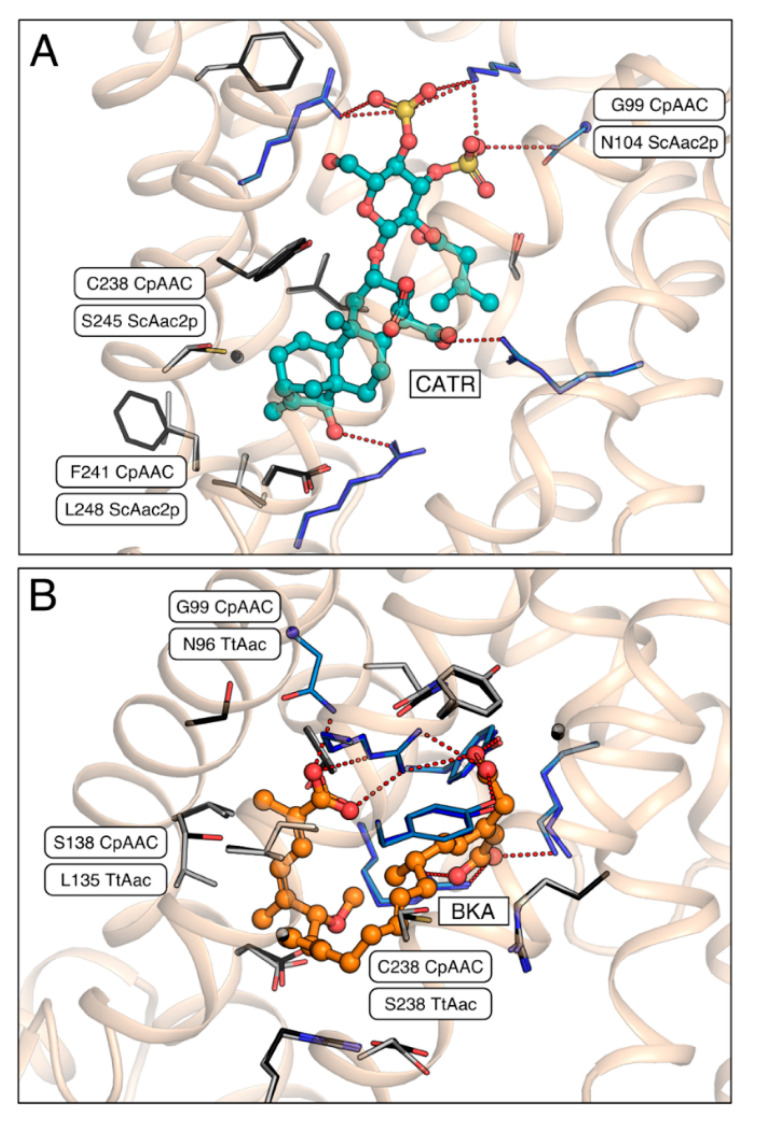
Carboxyatractyloside and bongkrekic acid binding to CpAAC. (**A**) Bound inhibitor carboxyatractyloside (CATR, teal) in the structure of ScAac2p (PDB entry 4c9g) [16] with the superimposed structural model of CpAAC, which was determined by SWISS-MODEL [38] using the ScAac2p structure (PDB entry 4c9g) as a template, following the alignment in Figure 2; (**B**) Bound inhibitor bongkrekic acid (BKA, orange) in the structure of TtAac (PDB entry 6gci) [28] with superimposed structural model of CpAAC, based on TtAac (PDB entry 6gci). All carriers are shown as a wheat cartoon with the inhibitors in ball-and-stick representations. Amino acid residues that form salt bridges or hydrogen bonds, or van der Waals interactions with the inhibitors, are shown in blue or grey carbon atoms stick representations, respectively. Interacting residues that are different between the structures and the CpAAC models are labeled.

**Figure 6 ijms-21-08971-f006:**
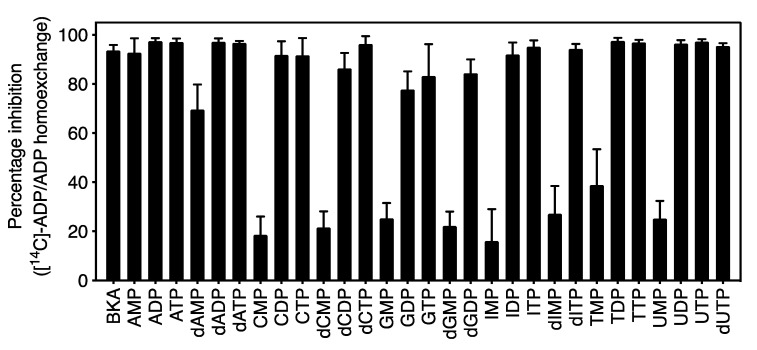
Nucleotide competition experiments of ADP/ADP homo-exchange by CpAAC. Percentage of inhibition of [^14^C]-ADP/ADP homo-exchange in the presence of 2.5 mM of non-radiolabeled nucleotides (1667-fold excess). Transport was initiated by the addition of 1.5 µM [^14^C]-ADP to fused vesicles, preloaded with 5 mM non-radiolabeled ADP. The data are represented by the mean and standard deviation of two biological repeats, each performed in quadruplicate.

**Figure 7 ijms-21-08971-f007:**
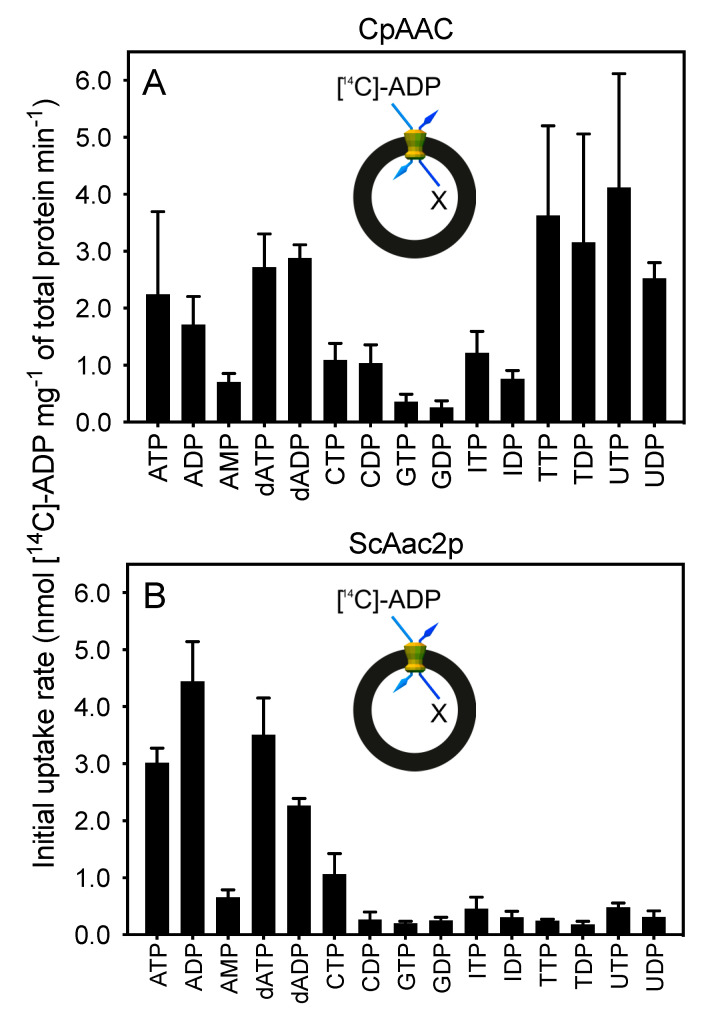
CpAAC has a broader substrate specificity than the yeast ADP/ATP carrier ScAac2p. Fused lactococcal membranes expressing (**A**) CpAAC or (**B**) ScAac2p were loaded with 2.5 mM of the indicated substrate (X), and hetero-exchange was initiated by the addition of 1.5 µM [^14^C]-ADP on the outside (see insets). The initial specific transport rates were determined over the linear part of the uptake curve, typically in the first 60 s. Data are represented by the mean and standard deviation of initial hetero-exchange rates of two biological repeats with six technical repeats in total.

**Figure 8 ijms-21-08971-f008:**
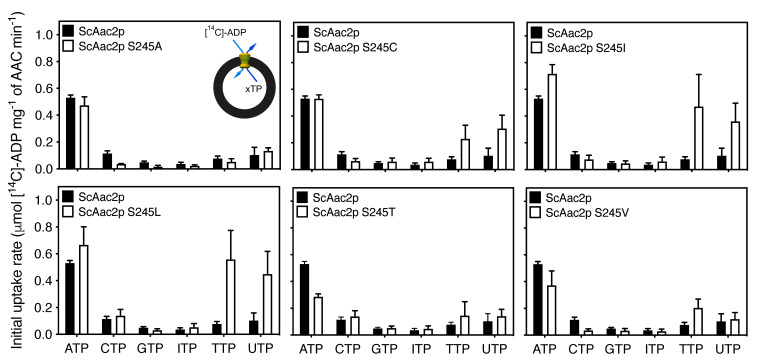
Nucleotide hetero-exchange by wild-type and S245 mutants of ScAac2p. Vesicles of lactococcal membranes expressing wild-type ScAac2p (black bars) or mutant carriers at position S245 (white bars) were loaded with 2.5 mM ATP, CTP, GTP, ITP, TTP, or UTP, and transport was initiated with the external addition of 1.5 µM [^14^C]-ADP. The specific initial transport rates were determined over the linear part of the uptake curve, typically in the first 60 s. Data are represented by the mean and standard deviation of two biological repeats with six technical repeats in total.

**Figure 9 ijms-21-08971-f009:**
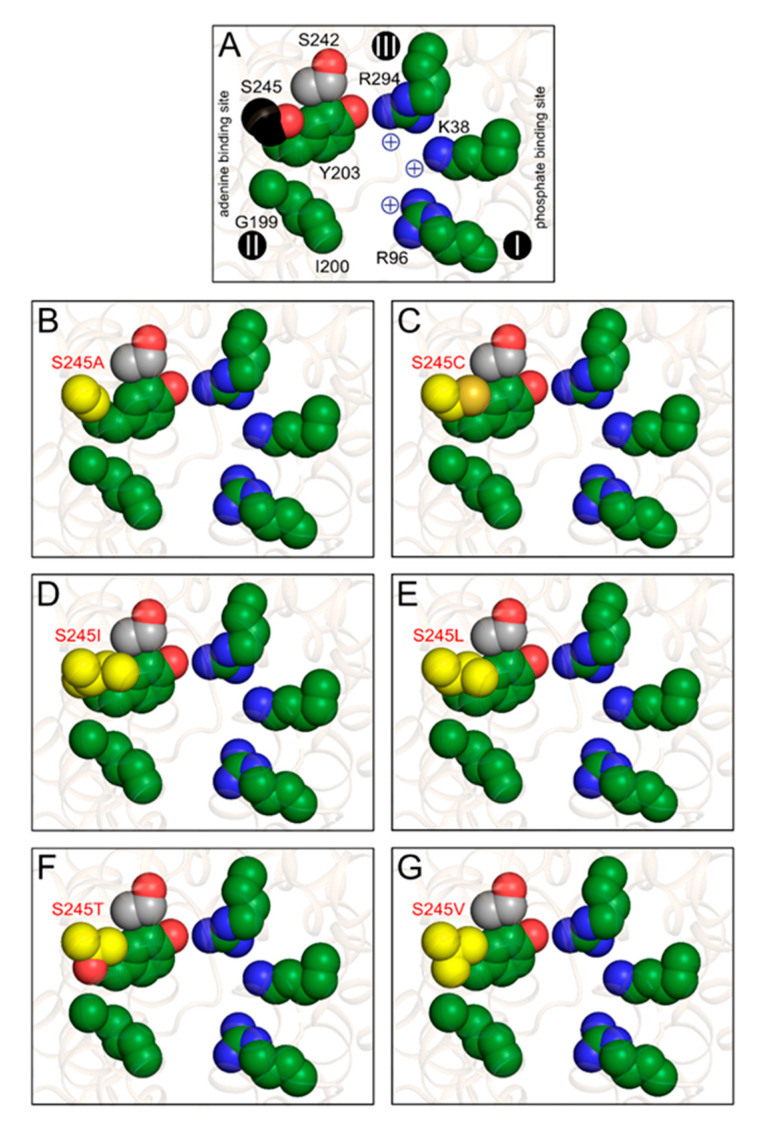
S245 substitutions change the properties of the putative substrate binding pocket. (**A**) Consensus substrate binding site of the yeast ADP/ATP carrier ScAac2p (PDB entry 4c9g) based on computational analysis [16,17,28,37]. The putative adenine binding site (G199, I200, and Y203) and positively charged residues (K38, R96, and R294) involved in binding of the phosphate moieties of the nucleotides are shown in green, whereas S242 and S245, which are close to this site, are shown in grey and black, respectively. The contact points I, II, an III on transmembrane helices H2, H4, and H6, respectively, are also indicated as black spheres with roman numerals [33,34]. Putative substrate binding site in (**A**) wild-type; (**B**) S245A; (**C**) S245C; (**D**) S245I; (**E**) S245L; (**F**) S245T; and (**G**) S245V with the substitutions, shown in yellow.

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
