# Peer review of "A Single Cysteine Residue in the Translocation Pathway of the Mitosomal ADP/ATP Carrier from Cryptosporidium parvum Confers a Broad Nucleotide Specificity"

_ijms, 2020, doi:10.3390/ijms21238971_

Round 1

Reviewer 1 Report

The manuscript ijms-989984 by King et al. deals with the characterization of a transport protein from the eukaryotic parasite Cryptosporidium parvum, namely of one member of the SLC25 mitochondrial carrier family belonging to the ADP/ATP carriers identified from the genome. The authors found that this member, called CpAAC, has a broader substrate specificity than mitochondrial orthologues. Moreover, they found a cysteine residue within the translocation pathway that differs from the usual serine and showed that this residue is responsible for the selectivity of the carrier. In fact, mutation of the original serine in the yeast carrier Aac2p, converted stringent ADP/ATP selectivity towards a broad substrate specificity. This study is interesting because the here described, clinically important, protist harbors a mitochondrion-related organelle, called mitosome, with a reduced set of transport proteins. Such transporters from the mitosomes are not yet well described, and the here presented findings give a molecular explanation for different substrate specificities.

Major Points:

(1) Identification of the molecular determinant of substrate specificity by the mutant study is very nice. However, taking the more selective yeast transporter to show broader substrate specificity when mutating the conserved serine (Figure 7), is interesting but raises the question, whether it works also in the other direction? Can you convert also the non-specific C. parvum transporter to a more selective one by introducing the serine instead of the cysteine?

(2) Figure 8 (in addition to Figure 7) adds information to the yeast carrier thus shifting the content of the manuscript, especially the Discussion part, more to the analysis of structure-function relationships within the yeast transporter what is not reflected by the Title and by the Abstract. This part should be either deleted/shortened or better be reflected by Title & Abstract.

(3) Introduction, Results, Discussion: The Authors published already another ADP/ATP transporter from the mitosome [37]. This should be more described in the Introduction, included in the Results section within Figure 2 & 3, and further discussed in comparison to the carrier described here within this study.

Minor Points:

(4) Title: write the species in italic ("Cryptosporidium parvum")

(5) Introduction, line 49; and also Results, line 104: The Authors mention the identification of 8 members of the SLC25 family within the genome of C. parvum, among them 1 putativ ADP/ATP carrier. What about the other 7 transport systems? Please give some more information, also to explain, why you have focalised here exclusively on the ADP/ATP carrier.

(6) Introduction, line 79: please explain " i analyses"?!

(7) Figure 3B, line 168: Explain the meaning of "initial rates".

(8) Figure 3C, line 171: Why is the human AAC (HsAAC1) shown?

(9) Results, lines 220-240: Reading would be more easy when the order of the presented Results corresponds between Figures and text (comment Fig. 5B before Fig. 6).

(10) Discussion, line 292: correct "Figure 6", the yeast carrier is not presented in Fig. 6 but in Figure 5 & 7.

(11) Materials and Methods, lines 388-395: Information about the construction of the yeast carrier ScAac2p is missing.

(12) Materials and Methods, lines 427-439: Analysis of the "initial transport rate" used in several figures should be explained.

(13) Materials and Methods: Structural modeling (Figure 2B, Figure 4) and computational analysis (used for Figure 8) is not explained.

Author Response

Reviewer 1

We thank the reviewer for their constructive comments, which have been very helpful in preparing a revised manuscript. We have extensively revised the manuscript accordingly and removed some minor typographical errors. Following is a point-by-point response to each of the comments:

The manuscript ijms-989984 by King et al. deals with the characterization of a transport protein from the eukaryotic parasite Cryptosporidium parvum, namely of one member of the SLC25 mitochondrial carrier family belonging to the ADP/ATP carriers identified from the genome. The authors found that this member, called CpAAC, has a broader substrate specificity than mitochondrial orthologues. Moreover, they found a cysteine residue within the translocation pathway that differs from the usual serine and showed that this residue is responsible for the selectivity of the carrier. In fact, mutation of the original serine in the yeast carrier Aac2p, converted stringent ADP/ATP selectivity towards a broad substrate specificity. This study is interesting because the here described, clinically important, protist harbors a mitochondrion-related organelle, called mitosome, with a reduced set of transport proteins. Such transporters from the mitosomes are not yet well described, and the here presented findings give a molecular explanation for different substrate specificities.

Major Points:

(1) Identification of the molecular determinant of substrate specificity by the mutant study is very nice. However, taking the more selective yeast transporter to show broader substrate specificity when mutating the conserved serine (Figure 7), is interesting but raises the question, whether it works also in the other direction? Can you convert also the non-specific C. parvum transporter to a more selective one by introducing the serine instead of the cysteine?

We agree that this experiment would be interesting to do, but we do not think it is essential as we have proven our hypothesis in different ways. We did not aim to study the molecular nature of the strict adenine nucleotide specificity of mitochondrial ADP/ATP carriers. Our goal, and the focus of the manuscript, is to characterise the mitosomal carrier CpAAC and explain its broad specificity by comparing it with other carrier homologues with stringent specificity. The reason for choosing ScAac2p is that we have an atomic structure, which allowed us to rationalise the changes introduced by the mutations. We show that the yeast ScAac2p, which shares the same binding site residues with CpAAC, can also be transformed to a carrier with the same broad nucleotide specificity by mutating Ser245 to cysteine. We have also shown that broadening of specificity is reproducible for specific hydrophobic substitutions, but not for other substitutions. Importantly, we also demonstrate that the adenine specificity is NOT related to the serine residue at that position, because when replaced by different residues the carrier can still transport adenine nucleotides. Thus, this residue is not directly in the binding site for adenine, which agrees with the structural analysis and everything else we know about substrate binding.

(2) Figure 8 (in addition to Figure 7) adds information to the yeast carrier thus shifting the content of the manuscript, especially the Discussion part, more to the analysis of structure-function relationships within the yeast transporter what is not reflected by the Title and by the Abstract. This part should be either deleted/shortened or better be reflected by Title & Abstract.

We feel that the abstract describes all the contents of the manuscript but we have made changes to the title and text, as suggested by the reviewer, retaining the focus of the manuscript on the broad substrate specificity of CpAAC, as that is the key message. The yeast work is only used to provide a molecular explanation for this particular effect. Luckily, the yeast binding site residues are identical, except for this one change, and therefore a good starting point, especially also because a structure is available.

(3) Introduction, Results, Discussion: The Authors published already another ADP/ATP transporter from the mitosome [37]. This should be more described in the Introduction, included in the Results section within Figure 2 & 3, and further discussed in comparison to the carrier described here within this study.

This is a very important point made by the reviewer. Interestingly, the Entamoeba histolytica mitosomal carrier also has a cysteine at an equivalent position, but the rest of the putative binding site residues are completely different, making a rationalisation of the substrate selectivity impossible. The Entamoeba carrier is also much smaller and has differences in the networks, showing that it is highly divergent. We have amended the text in Discussion to raise these points and have changed Figure 2 to include Entamoeba histolytica AAC. As a result, the figure became so large that it became necessary to make an additional Figure of the second panel.

Minor Points:

(4) Title: write the species in italic ("Cryptosporidium parvum")

This error was introduced by the formatting to the journal style, but we have corrected it.

(5) Introduction, line 49; and also Results, line 104: The Authors mention the identification of 8 members of the SLC25 family within the genome of C. parvum, among them 1 putative ADP/ATP carrier. What about the other 7 transport systems? Please give some more information, also to explain, why you have focalised here exclusively on the ADP/ATP carrier.

We have provided the requested information.

(6) Introduction, line 79: please explain " i analyses"?!

This error was introduced by the formatting to the journal style, but we have corrected it.

(7) Figure 3B, line 168: Explain the meaning of "initial rates".

Now explained this in the legend to Figure 4 by the new figure numbering.

(8) Figure 3C, line 171: Why is the human AAC (HsAAC1) shown?

HsAAC was our reference because the same expression system was used previously for HsAAc1, as now explained.

(9) Results, lines 220-240: Reading would be more easy when the order of the presented Results corresponds between Figures and text (comment Fig. 5B before Fig. 6).

In response, we have changed the figure order and panel order to improve the flow.

(10) Discussion, line 292: correct "Figure 6", the yeast carrier is not presented in Fig. 6 but in Figure 5 & 7.

This mistake has now been corrected.

(11) Materials and Methods, lines 388-395: Information about the construction of the yeast carrier ScAac2p is missing.

We have added the following line: The expression strains for the human ADP/ATP carrier HsAAC1 [27], and yeast ADP/ATP carrier [52, 53] have been described previously.

(12) Materials and Methods, lines 427-439: Analysis of the "initial transport rate" used in several figures should be explained.

We have altered the legend to Figure 4 to indicate the following: “The specific initial transport rates, when the applied chemical gradients of radio-labelled and cold substrates are maximal, are derived from the linear parts of the uptake curves, typically in the first minute, and corrected for the amount of protein”. We have also provided a better description in the other legends.

(13) Materials and Methods: Structural modeling (Figure 2B, Figure 4) and computational analysis (used for Figure 8) is not explained.

We have added this information in Materials and Methods and the legends to figures 3 and 5.

Reviewer 2 Report

The authors of manuscript entitled “ The Mitosomal ADP/ATP Carrier of 2 Cryptosporidium Parvum Has a Broader Nucleotide 3 Specificity than Its Mitochondrial Orthologs” have studied the transport properties of one member of the Cryptosporidium Parvum mitosomal carrier family. The alignement of C. parvum protein sequence to the sequences of yeast mitochondrial carrier family showed ADP/ATP carriers as the closet homologues. The CpAAC transporter expressed in lactococcal membranes showed a broader substrate specificity for nucleotides than ScAac2p. For the authors, this difference was determined by cysteine residue (C238) in C. parvum carrier which is serine residue (S245) in ScAac2p. In fact, when the serine residue was replaced by cysteine or hydrophobic residue in the Aac2p, the yeast carrier become able to transport other nucleotides than ADP and ATP.

Major revision:

1) The authors explain well why the cysteine 238 in CpAAC and the serine 245 in ScAac2p influence the substrate specificity, but to confirm their hypothesis the authors should measure the substrate specificity of CpAAC mutant C238S. Does this mutant acquire substrate selectivity like to that ScAac2p?

2) GDP and GTP inhibit homo-exchange ADP / ADP, but are not exchanged with ADP. For the authors this behavior is determined by a competition by GDP and GTP with the substrate (ADP). It is well known that GDP interact with UCP1, another member of mammalian SLC25A family, modulating its transport activity by allosteric interaction. Do the authors have experimental data (i.e. lineweaver-burk plot) about the GDP or GTP competition for substrate binding of CpAAC and/or ScAac2p?

Minor revision:

1) In the caption of figure 3C the primary antibody used should be indicated.

Author Response

We thank the reviewer for their constructive comments, which have been very helpful in preparing a revised manuscript. We have extensively revised the manuscript accordingly and removed some minor typographical errors. Following is a point-by-point response to each of the comments:

Reviewer 2

The authors of manuscript entitled " The Mitosomal ADP/ATP Carrier of 2 Cryptosporidium Parvum Has a Broader Nucleotide 3 Specificity than Its Mitochondrial Orthologs" have studied the transport properties of one member of the Cryptosporidium Parvum mitosomal carrier family. The alignment of C. parvum protein sequence to the sequences of yeast mitochondrial carrier family showed ADP/ATP carriers as the closet homologues. The CpAAC transporter expressed in lactococcal membranes showed a broader substrate specificity for nucleotides than ScAac2p. For the authors, this difference was determined by cysteine residue (C238) in C. parvum carrier which is serine residue (S245) in ScAac2p. In fact, when the serine residue was replaced by cysteine or hydrophobic residue in the Aac2p, the yeast carrier become able to transport other nucleotides than ADP and ATP.

Major revision:

1) The authors explain well why the cysteine 238 in CpAAC and the serine 245 in ScAac2p influence the substrate specificity, but to confirm their hypothesis the authors should measure the substrate specificity of CpAAC mutant C238S. Does this mutant acquire substrate selectivity like to that ScAac2p?

We agree that this experiment would be interesting to do, but we do not think it is essential as we have proven our hypothesis in different ways. We did not aim to study the molecular nature of the strict adenine nucleotide specificity of mitochondrial ADP/ATP carriers. Our goal, and the focus of the manuscript, is to characterise the mitosomal carrier CpAAC and explain its broad specificity by comparing it with other carrier homologues with stringent specificity. The reason for choosing ScAac2p is that we have an atomic structure, which allowed us to rationalise the changes introduced by the mutations with some accuracy. We show that the yeast ScAac2p, which shares the same binding site residues with CpAAC, can also be transformed to a carrier with the same broad nucleotide specificity by mutating Ser245 to cysteine. We have also shown that broadening of specificity is reproducible for specific hydrophobic substitutions, but not for other substitutions. Importantly, we also demonstrate that the adenine specificity is NOT related to the serine residue at that position, because when replaced by different residues the carrier can still transport adenine nucleotides. Thus, this residue is not directly in the binding site for adenine, which agrees with the structural analysis and everything else we know about substrate binding.

2) GDP and GTP inhibit homo-exchange ADP / ADP, but are not exchanged with ADP. For the authors this behavior is determined by a competition by GDP and GTP with the substrate (ADP). It is well known that GDP interact with UCP1, another member of mammalian SLC25A family, modulating its transport activity by allosteric interaction. Do the authors have experimental data (i.e. lineweaver-burk plot) about the GDP or GTP competition for substrate binding of CpAAC and/or ScAac2p?

Interesting point, but we have not done these experiments, as they do not directly address the claims in this paper. Mitochondrial carriers with a transport function are not known to have allosteric sites for substrates. UCP1 is an exception, but has a very different role, and the mechanism, which is not the subject of his paper, is still debated.

Minor revision:

1) In the caption of figure 3C the primary antibody used should be indicated.

We have now added the antibody information in the legend to figure 4, based on the new numbering.

Round 2

Reviewer 1 Report

The revised manuscript ijms-989984 by King et al. deals with the characterization of a transport protein from the eukaryotic parasite Cryptosporidium parvum, namely of one member of the SLC25 mitochondrial carrier family belonging to the ADP/ATP carriers identified from the genome. The authors found that this member, called CpAAC, has a broader substrate specificity than mitochondrial orthologues. Moreover, they found a cysteine residue within the translocation pathway that differs from the usual serine and showed that this residue is responsible for the selectivity of the carrier. In fact, mutation of the original serine in the yeast carrier Aac2p, converted stringent ADP/ATP selectivity towards a broad substrate specificity. This study is interesting because the here described, clinically important, protist harbors a mitochondrion-related organelle, called mitosome, with a reduced set of transport proteins. Such transporters from the mitosomes are not yet well described, and the here presented findings give a molecular explanation for different substrate specificities.

The Authors answered all the concerns raised during the first round of reviewing and revised their manuscript accordingly. Thus, the now submitted revised manuscript was corrected and improved.

Minor point:

(1) Correct "Cryptosporidium parvum" (line 225).

Reviewer 2 Report

The authors addressed all the concerns